# Clinical and Histopathological Features of Thyroid Cancer with *TERT* Promoter Molecular Alterations in Isolation Versus with Concurrent Molecular Alterations: A Multicenter Retrospective Study

**DOI:** 10.3390/cancers16203446

**Published:** 2024-10-11

**Authors:** Emily Steinberg, Orr Dimitstein, Grégoire B. Morand, Véronique-Isabelle Forest, Sabrina D. da Silva, Marc Pusztaszeri, Sama Alohali, Richard J. Payne

**Affiliations:** 1Faculty of Medicine and Health Sciences, McGill University, Montreal, QC H3A 0G4, Canada; 2Department of Otolaryngology—Head and Neck Surgery, Jewish General Hospital, Montreal, QC H3T 1E2, Canada; orr.dimitstein@mail.mcgill.ca (O.D.); gregoire.morand@mail.mcgill.ca (G.B.M.); veronique-isabe.forest@mcgill.ca (V.-I.F.); sama.alohali@mail.mcgill.ca (S.A.); richard.payne@mcgill.ca (R.J.P.); 3Department of Otolaryngology—Head and Neck Surgery, University Hospital Zurich, 8091 Zurich, Switzerland; 4Faculty of Medicine, University of Zurich, 8006 Zurich, Switzerland; 5Department of Otolaryngology—Head and Neck Surgery, McGill University Health Centre, Montreal, QC H4A 3J1, Canada; sabrina.wurzba@mcgill.ca; 6Department of Pathology, Jewish General Hospital, McGill University, Montreal, QC H3T 1E2, Canada; marc.pusztaszeri@mcgill.ca; 7Department of Otolaryngology—Head and Neck Surgery, King Faisal Specialist Hospital & Research Center, Riyadh 11564, Saudi Arabia

**Keywords:** thyroid cancer, molecular testing, cytology, TERTp, mutations, molecular alterations

## Abstract

**Simple Summary:**

This study explores the impact of telomerase reverse transcriptase (*TERTp*) molecular alterations on the behavior of thyroid nodules, exploring the differential behavior of these molecular alterations occurring alone versus with concurrent molecular alterations. By analyzing the data from thyroid cancer patients treated between 2017 and 2024, this study aimed to clarify how the genetic landscape of *TERTp* molecular alterations relates to severity and various clinical and histopathological features of disease. The study found that thyroid cancers harboring both *TERTp* and concurrent molecular alterations were more likely to be classified as high-risk and have aggressive histology in contrast to nodules with *TERTp* molecular alterations in isolation, which generally showed less aggressive behavior. These findings suggest that identifying concurrent molecular alterations in *TERTp*-positive thyroid nodules could improve cancer risk assessment and prognosis and inform more tailored treatment strategies.

**Abstract:**

**Background/Objectives:** Molecular testing of thyroid nodules enables the detection of genetic alterations, which can help assess the risk of malignancy and tumor behavior. While telomerase reverse transcriptase (*TERTp*) mutations are known to be associated with aggressive disease, their exact prognostic significance when occurring alone or with other molecular alterations remains underreported. **Methods:** This study examined patients with thyroid cancer treated at two tertiary care hospitals from 2017 to 2024. We compared tumor behavior in patients with *TERTp* molecular alterations occurring alone and with concurrent molecular alterations. Aggressive histologic subtypes were defined as tall-cell, hobnail, and columnar variants of papillary carcinoma, as well as poorly differentiated and anaplastic carcinoma. High-risk disease was defined according to the 2015 ATA guidelines as gross extrathyroidal extension, lymph node metastasis >3 cm, postoperative elevated serum thyroglobulin, distant metastases, and/or positive resection margins. Statistical analysis was performed to assess differences between groups. **Results:** 30 patients with *TERTp*-positive thyroid malignancies were included. *TERTp/BRAF V600E* was the most prevalent mutation combination (*n* = 13, 43.3%), followed by *TERTp* alone (*n* = 8, 26.7%) and *TERTp/RAS* (*n* = 7, 23.4%). *TERTp/EIF1AX/GNAS* and *TERTp/EIF1AX/PIK3CA* were the least common combinations (*n* = 1, 3.3% each). Nodules with *TERTp* and concurrent mutations were significantly more likely to be classified as high-risk (*p* = 0.006) and were more frequently associated with aggressive histologic subtypes (*p* = 0.003) compared to those with *TERTp* mutations alone, which tended to exhibit more benign behavior. **Conclusions:** Thyroid carcinomas harboring both *TERTp* and concurrent molecular alterations are associated with more aggressive features and a higher likelihood of being classified as high-risk. In contrast, *TERTp* mutations occurring alone do not confer an elevated risk.

## 1. Introduction

The incidence of thyroid cancer has increased by 300% worldwide over the last 30 years [1,2] and is projected to be in the top three most common cancers by 2040 [3]. Cancer registry studies found a significant increase not only in thyroid cancer overall but also in the incidence of advanced stages and aggressive subtypes [4]. Therefore, upon presentation with a thyroid nodule, it is paramount to provide accurate diagnostic and prognostic information for the patients and the treatment team. For this purpose, fine-needle aspiration cytology (FNAC) is now widely used for initial diagnosis of thyroid nodules. However, there are several limitations to FNA and cytopathology. FNAC requires a satisfactory sample of good technical quality, which is not always achieved and may be particularly difficult in smaller or deeper nodules. Also, FNAC allows for analysis of cells only from the region biopsied, which excludes potential malignant cells in other areas. FNAC is also unable to differentiate between a benign follicular adenoma and malignant follicular carcinoma [5,6,7]. The adjuvant of molecular testing provides additional context beyond the morphological information that cytopathology is based on. Several commercial tests are used for molecular testing of thyroid nodules, including ThyroSeq, ThyGeNEXT/ThyraMIR, and Afirma [8]. Common genetic alterations found in thyroid cancer include *BRAF V600E*, *NRAS*, *HRAS*, *KRAS*, *RET/PTC*, and *TERTp*.

Telomerase reverse transcriptase (*TERT*) is an enzymatic subunit of telomerase. Located on chromosome 5, *TERT* can undergo molecular alteration of its promoter region, leading to excessive telomere lengthening. This enables cancer cells to evade normal constraints on division and proliferation, causing cancer growth and metastasis [9,10]. *TERTp* genetic molecular alterations have been associated with vascular invasion, extrathyroidal extension, lymph node metastases, and tumor persistence/re-occurrence after treatment [9], with a relative 10-year survival rate 30.6% lower compared to wildtype *TERTp* [11].

A recent study explored the degree of aggressiveness of thyroid cancer in patients with *TERTp* molecular alterations compared to those with both *TERTp* and concurrent molecular alterations. It found that thyroid nodules with *TERTp* and concurrent molecular alterations were significantly more aggressive overall than *TERTp* molecular alterations alone [12]. However, the existing literature has not yet studied individual characteristics of tumor behavior, according to whether *TERTp* was occurring alone or with another molecular alteration(s). The relationship between the *TERTp* molecular profile and various features of disease is not yet known. This information could help provide more reliable prognostic factors to aid in decision making and treatment selection. The primary objective of this present study is to compare clinical and histopathological features, such as histologic pattern, nodal involvement, and distant metastases, of thyroid nodules with *TERTp* molecular alterations alone versus *TERTp* and concurrent molecular alterations.

## 2. Materials and Methods

### 2.1. Study Design

A multicenter retrospective cohort study was performed, evaluating patients of tertiary care hospitals who underwent thyroid surgery from January 2017 to January 2024 inclusively. This study included patients above 18 years old who underwent ultrasound-guided FNAC of dominant thyroid nodules and who were positive on molecular profile testing for *TERTp* molecular alterations. This study was granted approval by the Research Institute of the McGill University Health Centre, the McGill University Health Centre Research Ethics Board, and the CIUSSS West-Central Research Review Office.

### 2.2. FNAC and Molecular Testing

FNAC was performed as indicated by TIRADS guidelines [13]. For Bethesda III and IV nodules, options discussed included watchful waiting, molecular testing, or diagnostic lobectomy, according to the American Thyroid Association (ATA) guidelines [14]. In Bethesda V and VI nodules, molecular testing was performed as clinically indicated, particularly when the results would change management and/or extent of surgery. At the time of this study, molecular testing was not covered by insurance and was, therefore, paid for by the patients directly. No patient had any therapies or intervention for thyroid nodules prior to undergoing FNAC and subsequent molecular testing.

### 2.3. Definition of High Risk

As per the ATA risk stratification system, high-risk thyroid nodules were defined as the presence of at least one of the following: gross extrathyroidal extension (ETE), incomplete tumor resection (involved margins), distance metastases, postoperative elevated serum thyroglobulin levels suggestive of distant metastases, lymph node involvement with metastatic node ≥ 3 cm, and FTC with extensive vascular invasion [14]. In this study, serum thyroglobulin > 100 ng/mL was considered as thyroglobulin suggestive of distant metastases as defined by the ATA [12].

### 2.4. Definition of Aggressive Histology

In this study, aggressive histological subtypes were classified based on the ATA thyroid nodule guidelines as tall-cell, hobnail, and columnar subtypes of papillary thyroid carcinoma [14], as well as poorly differentiated and anaplastic carcinomas.

### 2.5. Data Collection

A total of 1902 charts from patients of the McGill University Health Center and the CIUSSS West-Central Jewish General Hospital were reviewed (Figure 1). Of these, 30 patients with *TERTp* molecular alterations were selected for inclusion in this study. Each patient’s electronic medical records were reviewed, and the following data were collected: patient demographics, Bethesda score, histopathological test results, imaging reports, laboratory values, molecular profile test results, and treatment. Pathology reports were reviewed for information on histological subtype, minimal or gross ETE, lymphovascular invasion (LVI), nodal involvement, and extra-nodal extension (ENE). Imaging was examined for presence of distant metastases or disease recurrence after treatment. Laboratory results were reviewed for postoperative serum thyroglobulin values. Possible treatments reviewed included surgery, radioactive iodine, external beam radiation, chemotherapy, and targeted treatments.

### 2.6. Data Analysis

Descriptive statistics of patient demographics and diagnostic classifications are presented with means and standard deviations. A chi-squared test was used to evaluate the association between categorical variables. A one-way ANOVA (with Fisher’s LSD adjustment for multiple comparisons) test was used to examine the difference of continuous variables. *p*-values ≤ 0.05 were considered statistically significant, and data analysis was carried out using STATA^®^ v17 (STATA Corp., College Station, TX, USA). Patients missing information for a given characteristic were excluded from the analysis for that particular variable; results were calculated based on the subset of patients with complete data for each characteristic individually.

## 3. Results

### 3.1. Participant Recruitment and Patient Demographics

The medical records of 1902 patients who underwent thyroid surgery were reviewed. Of these, 949 had undergone molecular testing by Afirma, ThyroSeq v3, or ThyGeNEXT. From this sample, 30 patients with *TERTp* molecular alterations were selected for inclusion in this study. Patient demographics are summarized in Table 1. Patient ages ranged from 45 to 86 years, with a mean age of 66.4 years (SD 11.3). The majority of patients included in this study were female (*n* = 23, 76.7%). There was no significant difference between groups for sex (*p* = 0.269).

### 3.2. Descriptive Statistics of TERTp Molecular Profile, Histopathological Features, and Treatment

Several molecular profiles were identified, including both single-gene *TERTp* molecular alterations as well as *TERTp* with concurrent molecular alterations (Figure 2). Those with *TERTp* molecular alterations were proved to be negative for any other known genetic alterations on molecular testing. *TERTp/BRAF V600E* was the most prevalent (*n* = 13, 43.3%), followed by *TERTp* alone (*n* = 8, 26.7%) and *TERTp/RAS* (*n* = 7, 23.4%). *TERTp/EIF1AX/GNAS* and *TERTp/EIF1AX/PIK3CA* were the least prevalent (*n* = 1, 3.3% each). The molecular profile of each nodule analyzed in this study and their associated characteristics are found in Table 2.

Final pathological assessment found that the majority of all included cases (*n* = 30) were diagnosed as malignant (*n* = 29, 96.7%). One (3.3%) patient with TERTp molecular alteration alone had a benign follicular adenoma.

The size of nodules varied significantly, ranging from 1.1 cm to 7.8 cm, with a mean size of 3.66 cm (SD 1.61). The number of involved lymph nodes observed also varied across patients, ranging from 0 to 11, with an average of 1.70 positive lymph nodes (SD 2.926). The most common histological subtypes were tall-cell (*n* = 14, 46.7%) and classical (*n* = 7, 23.3%) molecular alterations of papillary thyroid carcinoma.

Regarding treatment, surgical resection of the thyroid was performed in all 30 patients. Overall, 86.6% (*n* = 26) underwent total thyroidectomy, and 13.3% (*n* = 4) had hemithyroidectomy.

### 3.3. Association between TERTp Molecular Profile and Demographics and Nodule Size

*TERTp* alone was compared to *TERTp* with concurrent molecular alterations for several demographics and features of disease, and findings are summarized in Table 3. Further, a breakdown of these characteristics for each concurrent molecular alteration is found in Table 4, although statistical analysis comparing these subgroups was not significant due to small sample size and low statistical power. There is a significant difference in age between the two groups (*p* = 0.008). There was no significant difference found in thyroid nodule size (*p* = 0.341) or sex (*p* = 0.269) between the groups. Across both groups, the vast majority of nodules were found to be malignant on final postoperative pathology (87.5% of *TERTp* molecular alteration alone (*n* = 7) and 100% of *TERTp* with concurrent molecular alterations (*n* = 22)).

### 3.4. Association between TERTp Molecular Profile and Bethesda Classification

The Bethesda classification among this sample population ranged from Bethesda III to VI: Bethesda III (*n* = 5, 16.7%); Bethesda IV (*n* = 10, 33.3%); Bethesda V (*n* = 1, 2.2%); Bethesda VI (*n* = 14, 46.7%). Most patients with *TERTp* molecular alterations alone were classified as Bethesda IV (*n* = 5, 62.5%), while most with *TERTp* and concurrent molecular alterations were classified as Bethesda VI (*n* = 13, 59.1%). This difference in Bethesda classification was not found to be significant (*p* = 0.097).

Among patients with *TERTp* molecular alterations alone, two (25%) were classified as Bethesda III, five (62.5%) as Bethesda IV, none (0%) as Bethesda V, and one (12.5%) as Bethesda VI. Among patients with *TERTp* and concurrent molecular alterations, 3 (13.64%) were classified as Bethesda III, 5 (22.72%) as Bethesda IV, 1 (4.54%) as Bethesda V, and 13 (59.1%) as Bethesda IV. There was no significant difference found between groups regarding Bethesda classification (*p* = 0.097).

### 3.5. Effect of Molecular Profile on Risk Stratification

Sixteen nodules (53.3%) were classified as high-risk according to ATA risk stratification (see above) [11,12]. Of these, 15 (93.75%) had *TERTp* with concurrent molecular alterations, while the remaining 1 (6.25%) had *TERTp* molecular alteration alone. Of the 15 patients with *TERTp* and concurrent molecular alterations and high-risk nodules, 8 were classified as high-risk due to gross ETE, 4 due to incomplete resection (involved surgical margins), and 4 due to presence of distant metastases. Only one patient with *TERTp* molecular alteration alone was classified as high-risk. Statistical analysis revealed significant imbalance between the two groups (*TERTp* alone vs. *TERTp* with concurrent molecular alterations) (*p* = 0.006).

Of the 15 patients with *TERTp* and concurrent molecular alterations that were classified as high-risk, 11 (73.3%) had *TERTp/BRAF V600E* molecular alterations, while only 4 (26.7%) had *TERTp/RAS* molecular alterations. Also, 11 of 13 (84.6%) patients with *TERTp/BRAF V600E* molecular alterations were classified as high-risk, while 4 of 7 (57.1%) patients with *TERTp/RAS* molecular alterations were high-risk. Given the small sample size of each subgroup and low statistical power, statistical association could not be determined.

### 3.6. Association between TERTp Molecular Profile and Histological Subtype

Several histological subtypes were observed across both groups (Figure 3). Among eight patients with *TERTp* molecular alterations alone, nodules were distributed mainly between classical PTC (*n* = 4, 57.1%) and oncocytic (4N = 3, 2.8%) histologic subtypes, with a single case of a benign follicular adenoma. Among 22 nodules with *TERTp* and concurrent molecular alterations, the majority (*n* = 14, 63.6%) were classified as the tall-cell subtype of PTC. One patient had poorly differentiated carcinoma (*n* = 1), and the rest were distributed between classical PTC (*n* = 3, 13.6%), oncocytic (*n* = 2, 9.1%), and FTC (*n* = 2, 9.1%). Nodules with *TERTp/BRAF V600E* molecular alterations (*n* = 13) all displayed aggressive histology (*n* = 13, 100%), particularly the tall-cell subtype of PTC, while only one of seven (14.3%) nodules with *TERTp/RAS* molecular alterations had aggressive histology (tall-cell). Notably, no patients with *TERTp* molecular alterations alone had aggressive subtypes. Patients with *TERTp* and any concurrent molecular alterations were found to be significantly associated with aggressive pathologic subtypes compared to those with *TERTp* alone (*p* = 0.003).

### 3.7. Effect of Molecular Profile on Gross ETE

The association between gross ETE and *TERTp* molecular alteration status was examined. Among 21 cases without gross ETE, 38.1% (*n* = 8) exhibited *TERTp* molecular alterations alone, while 61.9% (*n* = 13) displayed *TERTp* molecular alterations alongside concurrent molecular alterations. Conversely, all cases with gross ETE had *TERTp* with concurrent molecular alterations. No nodules with *TERTp* concurrent mutations alone had gross ETE (*n* = 0). This association approached significance (*p* = 0.097).

### 3.8. Effect of Molecular Profile on Distant Metaseses

Six patients (20%) in this study had distant metastases. All these patients had *TERTp* with concurrent molecular alterations. Notably, no cases of distant metastases (*n* = 0) were observed among those with *TERTp* molecular alterations alone, though analysis did not reveal a statistically significant association between *TERTp* molecular status and distant metastases (*p* = 0.092).

## 4. Discussion

This study investigates the genetic landscape of thyroid malignancies, with a specific focus on *TERTp* molecular alterations occurring either alone or alongside other molecular alterations, such as *BRAF V600E*, *RAS*, *TERTp/EIF1AX/GNAS*, and *TERTp/EIF1AX/PIK3CA.* Our findings provide valuable insights into how these molecular profiles influence disease presentation. Consistent with previous research [12], we found that *TERTp* mutations with concurrent molecular alterations are more aggressive than *TERTp* mutations alone. Patients with these concurrent molecular alterations were more likely to be classified as high-risk according to the ATA guidelines (2015) and to exhibit aggressive histologic subtypes, including tall-cell PTC and poorly differentiated carcinoma.

Although the association between concurrent mutations and features like extrathyroidal extension (ETE) and distant metastases was not statistically significant, the trends observed suggest clinical relevance. These findings warrant further exploration to validate the potential impact of concurrent mutations on disease progression and to better understand the mechanisms driving thyroid cancer. Prior studies have demonstrated that coexisting *BRAF V600E* and *TERTp* mutations are linked to more aggressive clinical features, including larger tumors, ETE, and advanced disease stages [15,16,17]. Melo et al. found that *TERTp* molecular alterations were significantly associated with distant metastases in differentiated thyroid carcinomas [17]. However, this study did not consider the coexistence of other molecular alterations. Our study, however, found that patients with isolated *TERTp* mutations did not develop distant metastases, suggesting that *TERTp* mutations alone do not inherently promote cancer spread. This suggests that *TERTp* molecular alterations, when present without concurrent molecular alterations, do not inherently promote the spread of cancer to distant sites. Instead, it appears that the metastatic potential associated with *TERTp* that has been found in prior studies is likely due, in fact, to a dynamic interplay between *TERTp* and other molecular alterations. *TERTp* has been found to be an early marker of pre-malignant follicular adenomas, urothelial precursor lesions, and pre-neoplastic hepatic lesions [4,18,19]. *TERTp* molecular alterations, shown to be present in early, pre-malignant stages of disease, may be serve as a driving force for subsequent, more aggressive molecular alterations. This interaction, rather than *TERTp* molecular alteration in isolation, may be the underlying catalyst for the development of metastatic disease. This finding underscores the importance of considering the broader molecular context when assessing the metastatic risk of thyroid nodules with *TERTp* mutations.

These results may influence how we assess and manage patients with *TERTp*-positive thyroid cancer. While the ATA currently recommends pre-operative imaging for suspicious nodules, our study suggests that extensive imaging might be unnecessary for nodules with isolated *TERTp* mutations, given their less aggressive nature. Conversely, comprehensive preoperative staging might be crucial for patients with *TERTp* and concurrent molecular alterations, given their association with high-risk disease and metastasis.

Our analysis also highlighted differences between concurrent *TERTp/BRAF V600E* and *TERTp/RAS* molecular alterations. *TERTp/BRAF V600E* appeared to be more frequently associated with high-risk nodules and aggressive histologic subtypes compared to *TERTp/RAS*, which showed a less pronounced aggressive behavior. Concurrent molecular alterations in our population sample most frequently included *BRAF V600E* and *RAS*, which are the first and second most common molecular alterations found in thyroid FNA samples, respectively. Both alterations impact the mitogen-activated protein kinase (MAPK) pathway, inducing cell cycle progression [20]. *BRAF V600E* is associated with aggressive tumors with features of gross ETE, advanced stages, and metastases. They also more frequently present with recurrent or persistent disease postoperatively [20,21,22]. *RAS* (including *KRAS*, *NRAS*, and *HRAS*) molecular alterations are most frequently seen in follicular carcinomas and benign adenomas. *RAS* alterations are less frequently associated with features of aggressive disease, such as lymph node involvement and distant metastases [23]. Our findings align with prior research on different molecular alterations suggest that *TERTp/BRAF V600E* and *TERTp/RAS* molecular alterations may behave differently and should be studied separately in larger cohorts with greater statistical power to better understand their distinct impacts on thyroid cancer progression.

Several limitations of this study should be acknowledged. The retrospective design and non-randomized sampling may introduce biases, potentially affecting the accuracy and generalizability of the findings, as patients were selected from two tertiary care hospitals in Montreal, Canada. Additionally, environmental factors, particularly radiation exposure, play a significant role in thyroid cancer development [24], which may limit the applicability of our results to other populations. Further, some histological subtypes were not seen in our sample population, such as follicular variant of PTC. The study’s small sample size is consistent with the low prevalence of *TERTp* molecular alterations in thyroid cancer [25]; however, it still poses certain limitations, particularly for sub-group analyses and the statistical power of our findings. Future research with larger cohorts could further elucidate the differential impacts of various concurrent molecular alterations.

Despite these limitations, our study contributes to the growing body of literature on thyroid cancer, emphasizing the importance of molecular profiling in guiding clinical management and improving patient outcomes. Understanding the genetic drivers of thyroid cancer will enable the development of tailored therapeutic strategies, ultimately enhancing long-term outcomes for patients.

## 5. Conclusions

In this study, we explored the clinical and histopathological characteristics of thyroid nodules with *TERTp* molecular alterations alone versus those with concurrent molecular alterations, particularly *BRAF V600E* and *RAS*. Our findings provide new insights into the role of molecular profiles in disease presentation and risk stratification. We found that nodules with *TERTp* and concurrent molecular alterations are significantly more likely to be associated with aggressive histologic subtypes and classified as high-risk compared to those with *TERTp* mutations alone. While specific high-risk features, such as gross extrathyroidal extension (ETE) and distant metastases, did not reach statistical significance, the trends observed suggest potential clinical relevance. These findings underscore the critical importance of molecular profiling in guiding clinical management, tailoring therapeutic interventions, and ultimately improving patient outcomes.

## Figures and Tables

**Figure 1 cancers-16-03446-f001:**
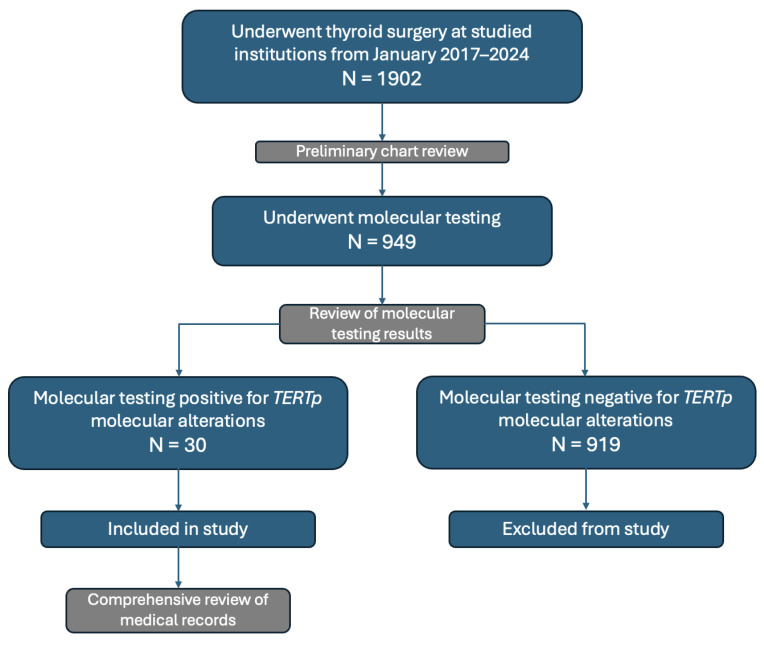
Selection of patients and data collection.

**Figure 2 cancers-16-03446-f002:**
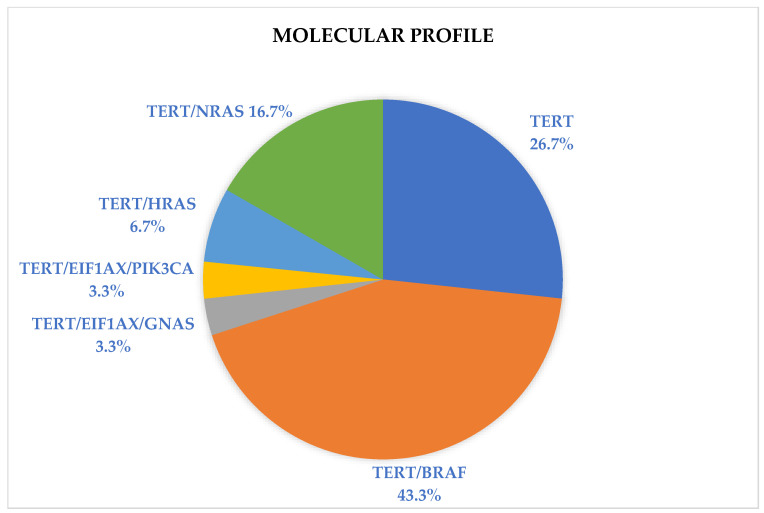
Molecular profile distribution.

**Figure 3 cancers-16-03446-f003:**
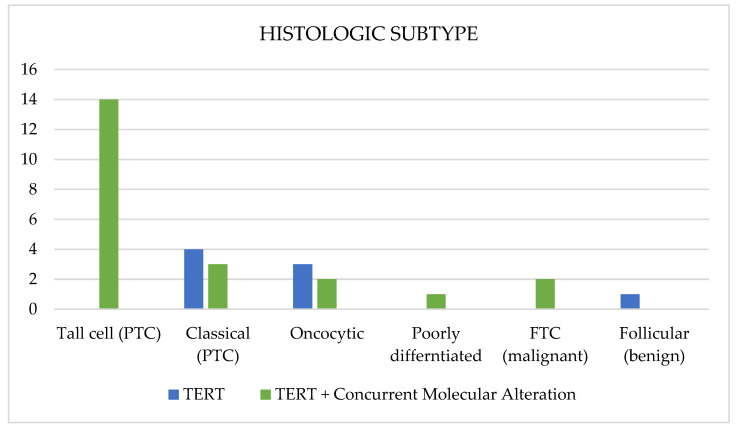
Histologic subtype distribution.

**Table 1 cancers-16-03446-t001:** Baseline demographics and nodule characteristics (*n* = 30).

		*n*	Frequency (%)
Age, mean		66.43	
Sex	Female	23	76.7
	Male	7	23.3
Histological subtype	Tall-cell (PTC)	14	46.7
	Classical (PTC)	7	23.3
	Follicular (benign)	1	3.3
	Follicular (FTC)	2	6.7
	Oncocytic	5	16.7
	Poorly differentiated	1	3.3
Molecular profile	*TERTp*	8	26.7
	*TERTp/BRAF*	13	43.3
	*TERTp/RAS*	7	23.4
	*TERTp/EIF1AX/GNAS*	1	3.3
	*TERTp/EIF1AX/PIK3CA*	1	3.3
Bethesda classification	Bethesda III	5	16.7
	Bethesda IV	10	33.3
	Bethesda V	1	3.3
	Bethesda VI	14	46.7
Positive lymph nodes, mean		1.70	
Nodule size, mean		3.663	
Final pathology	Benign	1	3.3
	Malignant	29	96.7
High risk	No	14	46.7
	Yes	16	53.3
Surgical procedure	Total thyroidectomy	26	86.7
	Hemi-thyroidectomy	4	13.3
Other treatment	Radioactive iodine	20	66.7
	External beam radiation	3	10
	Targeted treatment	3	10

**Table 2 cancers-16-03446-t002:** Molecular and histopathologic characteristics of nodules.

Molecular Profile	Age	Bethesda Classification	High Risk	Histology	Nodule Size (cm)	Positive Lymph Nodes
*TERTP*	75	IV	No	Follicular (benign)	2.6	0
*TERTP*	71	III	No	Classical PTC	4.6	0
*TERTP*	83	IV	No	Classical PTC	5	0
*TERTP*	82	III	Yes	Oncocytic	2.8	0
*TERTP*	75	IV	No	Classical PTC	2.3	0
*TERTP*	75	VI	No	Oncocytic	7.8	0
*TERTP*	82	IV	No	Oncocytic	4	0
*TERTP*	58	IV	No	Classical PTC	4	3
*TERTP/BRAF V600E*	68	VI	Yes	Tall-cell PTC	2.2	0
*TERTP/BRAF V600E*	78	VI	Yes	Tall-cell PTC	3.4	4
*TERTP/BRAF V600E*	73	VI	Yes	Tall-cell PTC	5.2	5
*TERTP/BRAF V600E*	53	VI	Yes	Tall-cell PTC	2.3	1
*TERTP/BRAF V600E*	67	VI	Yes	Tall-cell PTC	1.3	0
*TERTP/BRAF V600E*	56	VI	Yes	Tall-cell PTC	5	1
*TERTP/BRAF V600E*	70	V	Yes	Tall-cell PTC	5.6	2
*TERTP/BRAF V600E*	65	VI	Yes	Tall-cell PTC	1.9	2
*TERTP/BRAF V600E*	58	VI	Yes	Tall-cell PTC	2.8	1
*TERTP/BRAF V600E*	71	VI	Yes	Tall-cell PTC	3.5	1
*TERTP/BRAF V600E*	75	VI	Yes	Tall-cell PTC	2.6	10
*TERTP/BRAF V600E*	53	VI	No	Tall-cell PTC	2	11
*TERTP/BRAF V600E*	86	VI	No	Tall-cell PTC	1.1	0
*TERTP/HRAS*	53	IV	Yes	Tall-cell PTC	4	4
*TERTP/HRAS*	57	III	No	FTC	4.2	0
*TERTP/NRAS*	66	IV	No	Classical PTC	3.4	0
*TERTP/NRAS*	53	IV	Yes	Classical PTC	6	0
*TERTP/NRAS*	53	VI	Yes	Classical PTC	6.5	6
*TERTP/NRAS*	52	IV	Yes	Oncocytic	5.1	0
*TERTP/NRAS*	45	III	No	FTC	2	0
*TERTP/EIF1AX/GNAS*	64	IV	No	Poorly differentiated	2.7	0
*TERTP/EIF1AX/PIK3CA*	76	III	No	Oncocytic	4	0

**Table 3 cancers-16-03446-t003:** Clinical and histopathological characteristics of nodules with *TERTp* molecular alterations alone versus *TERTp* with concurrent molecular alterations.

		*TERTp* Alone (*n* = 8)	*TERTp* + Concurrent Molecular Alteration (*n* = 22)	Pearson Chi-Squared
Bethesda classification	BIII	2 (25.0%)	3 (13.6%)	*p* = 0.097
	BIV	5 (62.5%)	5 (22.7%)	
	BV	0 (0%)	1 (4.5%)	
	BVI	1 (12.5%)	13 (59.1%)	
Final pathology	Benign	1 (12.5%)	0 (0%)	*p* = 0.092
	Malignant	7 (87.5%)	22 (100%)	
High risk	No	7 (87.5%)	7 (31.82%)	*p* = 0.006
	Yes	1 (12.5%)	15 (68.18%)	
Aggressive histology	No	8 (100%)	7 (31.82%)	*p* = 0.003
	Yes	0 (0%)	15 (68.18%)	
Gross extrathyroidal extension	No	8 (100%)	13 (59.1%)	*p* = 0.097
	Yes	0 (0%)	8 (36.4%)	
	Not reported	0 (0%)	1 (4.5%)	
Distant metastases	No	7 (87.5%)	12 (54.5%)	*p* = 0.192
	Yes	0 (0%)	6 (27.3%)	
	Not reported	1 (12.5%)	4 (18.2%)	

Data are presented as observed frequencies. A Pearson chi-squared test was used to assess associations between variables, with a continuity correction applied where necessary. Results are considered significant at a level of *p* < 0.05.

**Table 4 cancers-16-03446-t004:** Clinical and histopathological characteristics of nodules with *TERTp* and different concurrent molecular alterations.

		*TERTp/BRAF* (*n* = 13)	*TERTp/RAS* (*n* = 7)	*TERTp/EIF1AX/GNAS* (*n* = 1)	*TERTp/EIF1AX/PIK3CA* (*n* = 1)
Bethesda classification	BIII	0 (0%)	2 (28.6%)	0 (0%)	1 (100%)
	BIV	0 (0%)	4 (57.1%)	1 (100%)	0 (0%)
	BV	1 (7.7%)	0 (0%)	0 (0%)	0 (0%)
	BVI	12 (92.3%)	1 (14.3%)	0 (0%)	0 (0%)
Final pathology	Tall-cell (PTC)	13 (100%)	1 (14.3%)	0 (0%)	0 (0%)
	Classical (PTC)	0 (0%)	3 (42.9%%)	0 (0%)	0 (0%)
	FTC	0 (0%)	2 (28.5%)	0 (0%)	0 (0%)
	Oncocytic	0 (0%)	1 (14.3%)	0 (0%)	1 (100%)
	Poorly differentiated	0 (0%)	0 (0%)	1 (100%)	0 (0%)
High risk	No	2 (15.4%)	3 (42.9%)	1 (100%)	1 (100%)
	Yes	11 (84.6%)	4 (57.1%)	0 (0%)	0 (0%)
Aggressive histology	No	0 (0%)	6 (85.7%)	0 (0%)	1 (100%)
	Yes	13 (100%)	1 (14.3%)	1 (100%)	0 (0%)
Gross Extrathyroidal extension	No	6 (46.15%)	5 (71.4%)	1 (100%)	1 (100%)
	Yes	6 (46.15%)	2 (28.6%)	0 (0%)	0 (0%)
	Not reported	1 (7.7%)	0 (0%)	0 (0%)	0 (0%)
Distant metastases	No	7 (53.8%)	4 (57.1%)	0 (0%)	1 (100%)
	Yes	3 (23.1%)	3 (42.9%)	0 (0%)	0 (0%)
	Not reported	3 (23.1%)	0 (0%)	1 (100%)	0 (0%)

## Data Availability

The data sets used and analyzed during this study are available from the corresponding author upon reasonable request.

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
