# Peer review of "Clinical and Histopathological Features of Thyroid Cancer with TERT Promoter Molecular Alterations in Isolation Versus with Concurrent Molecular Alterations: A Multicenter Retrospective Study"

_cancers, 2024, doi:10.3390/cancers16203446_

Round 1

Reviewer 1 Report

Comments and Suggestions for Authors

Interesting study and finding.

Since relative small number of follicular neoplasm -8cases,and half of them had TERT alone mutation. authors can make a table to describe the detail of those pathological findings -follicular and oncocytic.

Author Response

Comment: Since relative small number of follicular neoplasm -8cases,and half of them had TERT alone mutation. authors can make a table to describe the detail of those pathological findings -follicular and oncocytic.

Response: Thank you for this feedback! Of the 8 cases with follicular and oncocytic carcinoma, none had distant metastases or lymph node involvement and only one had macroscopic extra thyroidal extension. This is consistent with the finding that half had TERT mutations alone, which we see are not associated with distant mets and ETE. However, we believe a table highlighting this information is not as relevant to include in the manuscript given the main focus is the concurrent mutations, rather than histology. 

Reviewer 2 Report

Comments and Suggestions for Authors

Steinberg, et al.'s study, addresses an essential aspect of thyroid cancer prognostication by exploring the implications of TERT and other molecular mutations. Overall, the work is interesting and could have merit in the related field. However, several concerns should be addressed.

General comments

It is important for the authors to clearly define their rationale for considering the study multicenter (although they collected the samples from just two tertiary hospitals) and discuss any implications this may have on the study outcomes and generalizability in their manuscript.

The main concern is the limited sample size. Did the authors detect their study power? To be more informative, the authors should at least add “A pilot or preliminary study to the title.”

Introduction

- It would benefit from including more recent references (post-2018) to ensure the text is up-to-date, particularly regarding statistical data related to thyroid cancer incidence and molecular testing approaches.

- Line 65: please revise “Telomerase reverse transcriptase (TERT) is a protein subunit of telomerase”. I think the authors mean TERT coding for a protein ……etc!

- Elaborating on the limitations of FNAC in more detail could strengthen the argument for molecular testing.

Methods

- Lines 95 and 98: please provide supportive references for “according to the TIRADS guidelines, and “according to the TIRADS guidelines.”

- Can the authors clarify why they chose to adopt the earlier version of the ATA guidelines from 2015 in their study?

As part of the data transparency, the authors should provide a flowchart detailing the selection process, explaining how the initial 1,902 patient charts were narrowed down to the final 30 included in the study. This would enhance transparency and allow readers to understand the criteria used for selection more clearly.

- It would be beneficial for the authors to specify whether any patients in the studied cohort had prior exposure to the therapies mentioned in this section before sampling. And how was this considered in the analysis?

If any of the reviewed charts had incomplete data, the authors would need to discuss how they addressed potential missing data (incomplete records) in their analysis.

- Line 141: This could be better translocated to the methods section and supported with more detailed molecular work to cover all the mutations studied in this study, including the applied quality control measurements related to this issue.

Tables

The authors should provide a table footer that includes how the data were presented, the statistical test applied, and the significance level at which the results were considered significant.

Discussion

- line 249: “as high-risk according to the ATA guidelines.” The authors should provide the date of this guideline. Are the results in alignment with the recent one?

- The molecular basis for the differing prognoses associated with TERT promoter alterations, both in isolation and in conjunction with other molecular alterations, was unclear.

 Minor comment

The authors should replace “gender” with “sex,” as the latter refers to the normal biological difference between males and females.

-  It is essential to ensure that the abbreviations (ETE and LVI) are defined upon their first use in the text.

Author Response

Thank you so much for taking the time to review our manuscript and provide detailed feedback and suggestions. We hope our modifications clarify your concerns, and we are happy to provide further edits and clarifications as needed. 

Comment 1: It is important for the authors to clearly define their rationale for considering the study multicenter (although they collected the samples from just two tertiary hospitals) and discuss any implications this may have on the study outcomes and generalizability in their manuscript.

Response 1: the two hospitals from which we collected data are two large tertiary care centres, with differing patient demographics and social characteristics. For this reason, we consider the study multi-centre as it encompasses a good representation of the Montreal population. The limitation of this regarding generalizability of findings is mentioned in the discussion. 

Comment 2: The main concern is the limited sample size. Did the authors detect their study power? To be more informative, the authors should at least add “A pilot or preliminary study to the title.”

Response 2: We understand the concerns regarding the sample size. However, we would like to emphasize the uniqueness and specificity of our cohort, which consists of 30 patients carrying a rare mutation in TERT. Assembling such a cohort is particularly challenging due to the scarcity of this mutation in the general population.  Our dataset exhibits low variance, as we are working with a homogeneous group of patients, all of whom have a well-characterized mutation. This reduced variance lowers the standard error, leading to narrower confidence intervals and, consequently, enhanced statistical power, even with a smaller sample size. Additionally, we applied a one-tailed test for the statistical analysis, which is suitable given the clear direction of our hypothesis and gaining more statistical power. Although the sample size may seem small, the rarity of this mutation, coupled with the controlled variance and targeted hypothesis, makes our findings both valuable and robust.

Comment 3: It would benefit from including more recent references (post-2018) to ensure the text is up-to-date, particularly regarding statistical data related to thyroid cancer incidence and molecular testing approaches.

Response 3: Thank you for this feedback, older references (prior to 2018) for statistical data have been removed and replaced with more up-to-date findings. 

Comment 4: Line 65: please revise “Telomerase reverse transcriptase (TERT) is a protein subunit of telomerase”. I think the authors mean TERT coding for a protein ……etc!

Response 4: Wording was changes slightly, though TERT is described in several sources as a "subunit" of telomerase.

Comment 5: Elaborating on the limitations of FNAC in more detail could strengthen the argument for molecular testing.

Response 5: more information on the limitations of FNAC were added to the introduction. 

Comment 6: Lines 95 and 98: please provide supportive references for “according to the TIRADS guidelines, and “according to the TIRADS guidelines.”

Response 6: references were added. 

Comment 7:  Can the authors clarify why they chose to adopt the earlier version of the ATA guidelines from 2015 in their study?

Response 7: these guidelines (2015) are the most recent and commonly followed guidelines of their nature. While there have been newer guidelines from the ATA for more specific aspects of thyroid cancer such as anaplastic carcinoma, the 2015 guidelines are the most recent for general thyroid cancer overall.

Comment 8: As part of the data transparency, the authors should provide a flowchart detailing the selection process, explaining how the initial 1,902 patient charts were narrowed down to the final 30 included in the study. This would enhance transparency and allow readers to understand the criteria used for selection more clearly.

Response 8: a flowchart was created and added as a figure (Fig 1) under section 2.5 - data collection.

Comment 9: It would be beneficial for the authors to specify whether any patients in the studied cohort had prior exposure to the therapies mentioned in this section before sampling. And how was this considered in the analysis?

Response 9: patients had no prior exposure to therapies for thyroid cancer before sampling. This was added to the manuscript. 

Comment 10: If any of the reviewed charts had incomplete data, the authors would need to discuss how they addressed potential missing data (incomplete records) in their analysis.

Response 10: Percentages reported in the results were calculated based on the subset of patients with complete data for the relevant variables, ensuring accuracy and clarity. (i.e. percentages are calculated from cases with complete information). 

Comment 11: Line 141: This could be better translocated to the methods section and supported with more detailed molecular work to cover all the mutations studied in this study, including the applied quality control measurements related to this issue.

Response 11: this line already falls under methods

Comment 12: The authors should provide a table footer that includes how the data were presented, the statistical test applied, and the significance level at which the results were considered significant.

Response 12: footnote was added to table 3

Comment 13: line 249: “as high-risk according to the ATA guidelines.” The authors should provide the date of this guideline. Are the results in alignment with the recent one?

Response 13: the year was added. These results are the most recent guidelines. 

Comment 14: The molecular basis for the differing prognoses associated with TERT promoter alterations, both in isolation and in conjunction with other molecular alterations, was unclear.

Response 14: some more information on different mutations and suspected differing prognosis was added to the discussion 

Comment 15: The authors should replace “gender” with “sex,” as the latter refers to the normal biological difference between males and females.

Response 15: the text and tables contain the word "sex" 

Comment 16:  It is essential to ensure that the abbreviations (ETE and LVI) are defined upon their first use in the text.

Response 16: we verified that all abbreviations are properly and clearly defined. 

Reviewer 3 Report

Comments and Suggestions for Authors

The manuscript titled “Clinical and Histopathological Features of Thyroid Cancer with TERT Promoter Molecular Alterations in Isolation Versus with Concurrent Molecular Alterations: A Multicenter Retrospective 4 Study” of Emily Steinberg et al., reports an investigation about the incidence of TERT promoter molecular alterations occurring either alone or alongside other molecular alterations, such as BRAF V600E, RAS, TERT/EIF1AX/GNAS, and TERT/EIF1AX/PIK3CA in thyroid carcinomas.

The aim of this study was to investigate how the genetic landscape of TERT molecular alterations relates to severity and various clinical and histopathological features of PTC.

By multi-center retrospective cohort analysis, this investigation has selected 30 patients with TERT-positive thyroid malignancies.

Main contributions and strengths of this investigation can be found in having compared clinical and histopathological features, such as histologic pattern, nodal involvement, and distant metastases, of thyroid nodules with TERT promoter molecular alterations alone versus TERT and concurrent molecular alterations. Weakness is the number of patients, as the Authors have specified.

However, this paper is interesting and well-written; further, it meets the target of journal and may be accepted for publication.

Comments on the Quality of English Language

Good quality of English language

Author Response

Thank you for taking the time to review our manuscript and provide feedback. 

We understand the concerns regarding the sample size. However, we would like to emphasize the uniqueness and specificity of our cohort, which consists of 30 patients carrying a rare mutation in TERT. Assembling such a cohort is particularly challenging due to the scarcity of this mutation in the general population.  Our dataset exhibits low variance, as we are working with a homogeneous group of patients, all of whom have a well-characterized mutation. This reduced variance lowers the standard error, leading to narrower confidence intervals and, consequently, enhanced statistical power, even with a smaller sample size. Although the sample size may seem small, the rarity of this mutation, coupled with the controlled variance and targeted hypothesis, makes our findings both valuable and robust.

Reviewer 4 Report

Comments and Suggestions for Authors

A total of 1902 charts from patients of 2 tertiary hospitals were reviewed. Of these, who had Bethesda III -VI nodules, underwent molecular testing, and had subsequent surgery were included in this study. Eventually, 30 patients with TERT promoter (TERTp) molecular alterations were included in this study. The authors found that thyroid cancers harboring both TERTp and concurrent molecular alterations were classified as high-risk and have aggressive histology; in contrast, nodules with TERTp mutation alone generally showed less aggressive behavior. Thus, the authors suggest that identifying concurrent molecular alterations in TERTp-positive thyroid nodules could improve cancer risk assessment, prognosis, and inform more tailored treatment strategies.

Comments and suggestions

1.     Please abbreviate Telomerase reverse transcriptase promoter as TERTp in the manuscript.

2.     Since that the study was performed randomly by different molecular tests (ThyroSeq, ThyGeNEXT/ThyraMIR, and Afirma), please specify that the TERTp alone group was proved to be negative for other known genetic mutations or rearrangements specific for thyroid cancers even by different kits.

3.     Please express the histological subtype by papillary and follicular thyroid cancers by their subtypes, such as (Tall cell and classical, or follicular variant?), (minimally or widely invasive and Oncocytic), etc. in Tab. 1.

4.     Please specify tumor size and numbers as well as positive lymph nodes of each patient in Tab. 2 and determined their association with genetic profiles to complete the assessment by TNM and clinical (on counting age factor) stages.

5.     Please also specify whether the genetic profile was associated with radioactive iodine uptake in this study.

6.     Please specify the patient with final diagnosis of follicular adenoma (FA) in Tab. 2. Please also remind the readers that TERTp mutation may serve as an early genetic event activating telomerase in FA and atypical FA (Cancer 2014;120: 2965-79).

7.     Please describe the preoperative FNAC report of the Bethesda VI in one of the 5 oncocytic cancers.

8.     Please specify the association of each concurrent (BRAF, RAS and E1F1AX) mutations with the clinical findings in Tab. 3.

Author Response

Thank you for your feedback! your suggestions will certainly enhance the quality of our paper and we appreciate you taking the time to review our manuscript. 

Comment 1: Please abbreviate Telomerase reverse transcriptase promoter as TERTp in the manuscript.

Response 1: this change was made throughout the manuscript

Comment 2: Since that the study was performed randomly by different molecular tests (ThyroSeq, ThyGeNEXT/ThyraMIR, and Afirma), please specify that the TERTp alone group was proved to be negative for other known genetic mutations or rearrangements specific for thyroid cancers even by different kits.

Response 2: this was added to section 3.2 

Comment 3: Please express the histological subtype by papillary and follicular thyroid cancers by their subtypes, such as (Tall cell and classical, or follicular variant?), (minimally or widely invasive and Oncocytic), etc. in Tab. 1.

Response 3: classical and tall cell variants of PTC was clarified. Follicular was divided into benign adenoma and malignant FTC. 

Comment 4: Please specify tumor size and numbers as well as positive lymph nodes of each patient in Tab. 2 and determined their association with genetic profiles to complete the assessment by TNM and clinical (on counting age factor) stages.

Response 4: this information was added to the table. 

Comment 5: Please also specify whether the genetic profile was associated with radioactive iodine uptake in this study.

Response 5: information on RAI uptake was not included in this study as it was not widely available in the files of many patients. 

Comment 6: Please specify the patient with final diagnosis of follicular adenoma (FA) in Tab. 2. Please also remind the readers that TERTp mutation may serve as an early genetic event activating telomerase in FA and atypical FA (Cancer 2014;120: 2965-79).

Response 6: the follicular adenoma case was highlighted, and this reference was added to the discussion. 

Comment 7: Please describe the preoperative FNAC report of the Bethesda VI in one of the 5 oncocytic cancers.

Response 7: This patient was initially thought to have PTC on initial FNAC, though on final pathology post-operatively, he was officially diagnosed with hurthle cell (oncocytic) carcinoma. 

Comment 8: Please specify the association of each concurrent (BRAF, RAS and E1F1AX) mutations with the clinical findings in Tab. 3.

Response 8: A separate table (Tab 4) was created to highlight the characteristics of each specific concurrent mutation. 

Round 2

Reviewer 2 Report

Comments and Suggestions for Authors

Thanks to the authors for addressing the raised concerns. Before the final publication, please provide a supportive citation for the newly added text (lines 59-65) in the introduction section. 

Author Response

Comment 1: Before the final publication, please provide a supportive citation for the newly added text (lines 59-65) in the introduction section.

Response 1: thank you for your feedback. References have been added for this added text on limitations FNAC

Reviewer 4 Report

Comments and Suggestions for Authors

1.      Please specify the final pathology in detail as classical PTC (cPTC), tall cell PTC (tcPTC), FTC, oncocytic, or PDTC instead of benign or malignant in Tab. 4.

2.      Please also address that the follicular variant of PTC (FVPTC, the less aggressive subtype of PTC may harbor RAS as well as BRAF mutations) was not included in this study in the limitation of the study.

Author Response

Comment 1: Please specify the final pathology in detail as classical PTC (cPTC), tall cell PTC (tcPTC), FTC, oncocytic, or PDTC instead of benign or malignant in Tab. 4.

Response 1: this was changed in table 4. Additionally, the subtypes were classified accordingly in table 2 as well. 

Comment 2: Please also address that the follicular variant of PTC (FVPTC, the less aggressive subtype of PTC may harbor RAS as well as BRAF mutations) was not included in this study in the limitation of the study.

Response 2: this was added to the discussion section as part of the study limitations.